# Secure Service Composition with Quantitative Information Flow Evaluation in Mobile Computing Environments

**DOI:** 10.3390/e21080753

**Published:** 2019-08-01

**Authors:** Ning Xi, Jing Lv, Cong Sun, Jianfeng Ma

**Affiliations:** School of Cyber Engineering, Xidian University, Xi’an 710071, China

**Keywords:** quantitative information flow, secure information flow model, service composition, mobile computing

## Abstract

The advances in mobile technologies enable mobile devices to cooperate with each other to perform complex tasks to satisfy users’ composite service requirements. However, data with different sensitivities and heterogeneous systems with diverse security policies pose a great challenge on information flow security during the service composition across multiple mobile devices. The qualitative information flow control mechanism based on non-interference provides a solid security assurance on the propagation of customer’s private data across multiple service participants. However, strict discipline limits the service availability and may cause a high failure rate on service composition. Therefore, we propose a distributed quantitative information flow evaluation approach for service composition across multiple devices in mobile environments. The quantitative approach provides us a more precise way to evaluate the leakage and supports the customized disciplines on information flow security for the diverse requirements of different customers. Considering the limited energy feature on mobile devices, we use a distributed evaluation approach to provide a better balance on consumption on each service participant. Through the experiments and evaluations, the results indicate that our approach can improve the availability of composite service effectively while the security can be ensured.

## 1. Introduction

With the development of intelligent terminal, 5G and IoT technologies, various mobile applications enrich our daily lives with more flexible and convenient IT services delivery [1,2]. Moreover, high speed processors and stable connections enable the efficient service interactions among different mobile devices. Based on service-oriented architecture, service composition across multiple mobile devices provides a promising way for integrating several distributed services to satisfy users’ complex requirements [3]. Most works focus on improving the efficiency and availability of composite services in mobile computing [4,5,6]. However, various data with different sensitivities and heterogeneous systems with diverse security policies pose a great challenge on information flow security during the service composition across multiple devices [7]. In particular, if one service component contains malicious code or vulnerabilities, customers’ sensitive data may be leaked. In addition, illegal providers or attackers may collude together to eavesdrop private data more effectively based on feedback from different components, in which private data may also be leaked even if individual service is protected by an access control mechanism [8,9].

In order to prevent the data leakage during service composition, types of information flow mechanisms are proposed including type of system [10], model checking [11,12], program static analysis [13,14] and real-time monitoring [15,16]. Considering the limited energy and dynamic composition relationships, we propose a distributed information flow verification framework for secure service composition in mobile computing environments [7]. Although these approaches provide a solid assurance on information flow security of composite service, implementing them in a real application is still a challenge. These approaches are based on a qualitative discipline, i.e., non-interference [17], which strictly limits the complete absence of any causal flow from high-level sources to low-level sinks. Too strict discipline causes the loss of service availability on account of the security limitations on the cross-level operations in program. It may also cause a high failure rate on service composition because few services can satisfy the discipline. In fact, it is usually permitted in practice to tolerate some leakage for a better service availability. For example, the area of our location may be allowed to be observed by mobile service providers for a customized and more precise route planning. Therefore, for a better balance on service security and availability, it is important for us to measure “how much” information is leaked and “how many” leaks are allowed by customers during the service composition.

In order to quantifying the leakage, many quantitative information flow approaches are proposed based on Shannon’s information theory [18]. The authors in [19,20] propose the approach to quantify interference in a simple imperative language for the information flow verification. The authors in [21] present an automatic method for information-flow verification that discovers what information is leaked and computes its comprehensive quantitative interpretation. The authors in [22] establish a tight bound on the maximum leakage from repeated independent runs. However, these approaches mainly focus on a single program that works in a centralized way. During the service composition, there may be several services with similar functions but developed by different mobile service providers, which requires us to select appropriate services for optimized performance [6,23]. It would be a resource-consuming work to evaluate all possible services by a single piece of equipment, which is hard to be implemented due to the energy-limited features of mobile terminals. In addition, all candidate services must be reevaluated even if a small change of one service occurs, which also increases the evaluation load on mobile devices.

In this paper, we present a distributed quantitative information flow evaluation approach applied on the service composition in a mobile computing environment. Our contributions mainly include: (1) we make the quantifying rules on information flows based on the static analysis; (2) we propose the quantitative definition on secure information flow in composite services and specify the security constraints on each service component for distributed evaluation; (3) we design a distributed quantitative information flow evaluation framework and approach for secure service composition in mobile environment, which can provide a better service availability and load balance with affordable costs.

The rest of the paper is structured as follows. Section 2 presents the basic models of the mobile service system. Section 3 details the quantifying rules and the security theorems based on the static analysis. In Section 4, the distributed quantitative information flow evaluation approach is proposed according to the security theorems. Section 5 evaluates our proposed approach. Section 6 concludes the paper.

## 2. Mobile Service System

### 2.1. System Model

As shown in Figure 1, the Mobile Service System (MSS) is a distributed IT system consisting of multiple network domains. A domain can have various types of resources, such as data, information, and other physical resources. Mobile terminals in the domain can use these resources and its application functions to provide various services to users, e.g., $s_{1}$, $s_{2}$, etc. These services can be composed together for a more complex users’ application. Moreover, there are several candidate services that can execute the similar functions for a given service. These services can be developed by different service providers. For example, $s_{1}$ can be provided by *A*, *B* or other service providers, i.e., $s_{1|A}$, $s_{1|B}$ and so on. In addition, there is also a security authority in each domain for the security management in the domain.

Referring to the system model in [7], each domain *D* can be represented as D=〈S,R,SA〉, where *S* is the set of various services, i.e., $S=\{s_{0},s_{1},\ldots \}$; *R* is the set of physical resources that can be collected by mobile services in the domain, e.g., environment data, traffic data and so on; SA is the security authority. Each service $s_{i}$ in *S* is defined as a tuple $s_{i}=\langle id_{i},dom_{i},In_{i},Out_{i},Pg_{i},Ce_{i}\rangle$, where $id_{i}$ is the identifier of the service provider; $dom_{i}$ is the domain that $s_{i}$ belongs to; $In_{i}$ is the set of inputs in $s_{i}$; $Out_{i}$ is the set of the outputs in $s_{i}$; $Pg_{i}$ is the program of $s_{i}$, which describes the execution procedure of $s_{i}$; $Ce_{i}$ is the certificate of the service which specifies the security properties.

Due to the user’s complex requirements, different services $s_{i}$ in multiple domains may be composed together to achieve the service goal. In this paper, we investigate a typical composite service, i.e., the service chain $S_{ch}$ [7], as shown in Figure 2. A service chain is widely used in service composition because of its simplified composition structure which is easy to deploy and control. In service chain $S_{ch}$, $s_{0}$ receives the request from user and starts the composition procedure. Then, each service $s_{i}$, 0<i<n, receives the intermediate result as the inputs from its unique predecessor $s_{i-1}$, executes the service program $Pg_{i}$ and outputs the intermediate result to its unique successor $s_{i+1}$. Finally, the last one $s_{n}$ sends the final results to the user. During the execution, service providers (SP) can input and obtain some data according to the service request, which may cause the leakage of a user’s private information.

### 2.2. Threat Model

Based on our system model, we make the following assumptions about the security capabilities of the participants in MSS.

• **User**: The user is the data owner who has access to all inputs and outputs including public and private information. We partition them into two sets: *L*(low) for public data and *H*(high) for private data. In addition, users don’t intentionally collude to leak the private data.

• **Service Providers**: Service providers from different mobile devices are honest but curious. They execute the service functions in accordance with their descriptions. They can not access users’ high-level data directly due to the privacy policies, but they can freely observe the public data including all low-level inputs and outputs before and after (but not during) the service’s execution. Some of them may try to analyze the value of the private data based on users’ low-level inputs and outputs on purpose. In addition, different services may collude together to analyze a user’s private data more effectively.

For a clear description, we define $LIn_{i}$ and $LOut_{i}$ as the public inputs and outputs with low-level security *L*. $HIn_{i}$ and $HOut_{i}$ are defined as the private inputs and outputs with high-level security *H*. Then, we can obtain that $In_{i}=LIn_{i}⋃HIn_{i}$ and $Out_{i}=LOut_{i}⋃HOut_{i}$.

• **SA**: Security Authority is the trusted third party that executes the security function honestly without any interception and manipulation.

## 3. Quantitative Information Flow Model for Service Composition in a Mobile Computing Environment

### 3.1. Quantitative Information Flow Model Based on Information Theory

Shannon’s theory provides a standard measurement on information quantity known as self-information or entropy. For random variable *X* for storing different data x∈X in service program $Pg_{i}$, its entropy can be defined as [24]
(1)$$H\left(X\right)=\sum_{x}p\left(x\right)\log\frac{1}{p(x)},$$
where *X* is a random variable, p(x) is shorthand for P(X=x), which is the probability of X=x, and the sum is over the range of *X*. In Equation (Equation 1), the base for log is conventional to use base 2 for the analysis in computer program.

The conditional entropy can be used to represent the amount of information carried by *X* give the knowledge of variable *Y*, which is defined as:(2)$$H\left(X|Y\right)=\sum_{y}p\left(y\right)H\left(X|Y=y\right),$$
where $H\left(X|Y=y\right)=\sum_{x}p\left(x|y\right)\log\frac{1}{p(x|y)}$, and p(x|y) is the probability that random variable X=x given that random variable Y=y.

Based on the information quantity on each variable, the mutual information provides a general way of measuring the amount of information stored by *X* that can be learned by observing another random variable *Y*, which is defined as: (3)$$\begin{array}{c} I\left(X;Y\right)=\sum_{x}\sum_{y}p\left(x,y\right)\log\frac{p(x,y)}{p(x)p(y)}\\ \;\;=H(X)+H(Y)-H(X,Y)\\ \;\;=H(X)-H(X|Y)\\ \;\;=H(Y)-H(Y|X). \end{array}$$

According to our system model, LIn and LOut are the low-level inputs and outputs that service providers can observe during the service composition. Based on Equation (Equation 3), for each X∈In, the leakage through the flow from *X* to Y∈Out can be defined as
(4)$$F_{LIn}\left(X⇝Y\right)=I\left(X;YlIn\right)=H\left(XlIn\right)-H\left(X|Y,LIn\right).$$

We use $F_{LIn}\left(X⇝LOut\right)$ to represent the overall leakage of *X* through all different flows from *X* to any *Y* in LOut. For a clear description, we assume all the inputs and outputs are *k* bits variables and the inputs are uniformly distributed and independent from each other in the following calculation. Then, we can derive the first basic Quantifying Rule (**QR**) as follows:

**QR** **1.**
*∀X∈In, $Max(F_{LIn}\left(X⇝LOut\right))=H\left(X\right)=k$ and $Min(F_{LIn}\left(X⇝LOut\right))=0$.*


Then, we can obtain the quantitative definition on information flow security in a service as follows:

**Definition** **1.**
*∀X∈HIn in a service, the flows in service are K-secure, for 0≤K≤k, if*
$$F_{LIn}\left(X⇝LOut\right)<K,$$
*where LIn and LOut are the low-level input and output observations, and K is the security threshold that depends on a user’s requirement and the system running environment.*


According to Definition 1, we can derive the following two facts: (1) if K=0, it requires that there is no flow from *X* to any *Y* in LOut, which becomes the qualitative definition of standard non-interference as shown in [25]. (2) if 0<K≤k, it is considered secure if there is at least k−K unknown bits for service providers. A user can choose different thresholds for diverse security requirements in different running environments.

### 3.2. Quantifying the Information Flow in Service Components

Clark et al. [20] propose the basic analysis rules to quantify leakage for a software program with sequence, branch and loop structures. These rules are specified for the single program analysis in a centralized way, which aim at quantifying the overall leakage instead of each flow’s leakage. Therefore, they don’t support distributed quantifying across multiple services. Based on some basic rules in [20], we design the improved quantifying rules on different information flows through the static analysis. Our rules are based on the worst case assumption for guaranteeing the security. First, for the $s_{i}$’s program $Pg_{i}$, its syntax can be defined as follow by referring to [20]. It is a simple imperative language including all basic notions, operations and structures in a program:$$\begin{array}{c} C\in Com\;var\in Var\;E\in Exp\;B\in BExp\;const\in N\\ P::=P;C\;|\;C\\ C::=\mathrm{skip}\;|\;var:=E\\ \;\;|\;\mathrm{if}\left(B\right)\;\mathrm{then}\;C\;\mathrm{else}\;C^{\prime }\\ \;\;|\;\mathrm{while}(B)\;C\\ E::=var\;|\;const\;|\;E+E^{\prime }\;|\;E*E^{\prime }\\ B::=E\;R\;E^{\prime }\;|\;\neg B\;|\;B\wedge B^{\prime }\;|\;B\vee \;B^{\prime }\\ R::=<\;|\;>\;|\;==. \end{array}$$

There are two kinds of flows to consider, i.e., explicit and implicit flow [17]. The explicit flow occurs as a result of executing the assignment statement. For example, for the statement $var^{\prime }:=E$, if *E* contains variable var, there is an explicit flow between var and $var^{\prime }$, namely $var\to var^{\prime }$. Implicit flow occurs as a result of executing a statement *C* or not when this statement *C* is conditioned on the value of an binary expression *B*. This type of flow usually exists in the branch and the loop structures. If *B* contains var and $var^{\prime }$ appears in *C* or $C^{\prime }$ as an objective variable, there is an implicit flow between var and $var^{\prime }$, namely $var\overset{B}{⤏}var^{\prime }$. Based on the basic dependence and its transitivity, we can define the intra flows from var to $var^{\prime }$ as follow, which is represented as $\delta (var,var^{\prime })$.

**Definition** **2.**
*$\forall var,var^{\prime }\in Var$ in $s_{i}$, and there are four cases to consider:*

*(1) ∃v∈Var that satisfies var→v and $v\to var^{\prime }$, then $\delta \left(var,var^{\prime }\right)=var\to var^{\prime }$.*

*(2) ∃v∈Var that satisfies var→v and $v\overset{B}{⤏}var^{\prime }$, then $\delta \left(var,var^{\prime }\right)=var\overset{B}{⤏}var^{\prime }$.*

*(3) ∃v∈Var that satisfies $var\overset{B}{⤏}v$ and $v\to var^{\prime }$, then $\delta \left(var,var^{\prime }\right)=var\overset{B}{⤏}var^{\prime }$.*

*(4) ∃v∈Var that satisfies $var\overset{B}{⤏}v$ and $v\overset{B^{\prime }}{⤏}var^{\prime }$, then $\delta \left(var,var^{\prime }\right)=var\overset{B}{⤏}var^{\prime }$.*


Here, we use $var⇝var^{\prime }$ to represent all the flows from var to $var^{\prime }$. According to our attacker model, the attacker can observe the low-level inputs and outputs before and after the execution of service. Thus, we mainly focus on the flows between the inputs and outputs. In addition, we can also obtain that $In_{i}\subseteq Var$ and $Out_{i}\subseteq Var$. In order to analyze these flows among the different inputs and outputs in $s_{i}$, we construct the PDG (Program Dependence Graph) first [26], and then use the program slicing [27] to obtain all the flows from inputs to outputs based on Definition 2. Then, we define $F_{i}=\left\{\delta \left(X,Y\right)|X\in In_{i},Y\in Out_{i}\right\}$ as the set of all intra flows in $s_{i}$ for the following calculation.

Based on the definition of $F_{i}$, we can derive the following quantifying rules on the leakage of each flow in $s_{i}$. In this paper, we consider the worst case assumptions in which we focus on the computation of the upper bound of leakage for a strong security assurance.

**QR** **2.**
*$\forall X\in HIn_{i}$ and $Y\in LOut_{i}$ satisfy X→Y, then we have*
(5)$$F_{LIn_{i}}\left(X\to Y\right)=H\left(X\right)=k,$$
*where $LIn_{i}$ is the low level input observations in $s_{i}$.*


**QR** **3.**
*$\forall X\in HIn_{i}$ and $Y\in LOut_{i}$ satisfy $X\overset{B}{⤏}Y$, then we have*
(6)$$F_{LIn_{i}}\left(X\overset{B}{⤏}Y\right)=\left\{\begin{array}{cc} 1 & B::=E<E^{\prime }\;|\;E>E^{\prime },\\ F^{Eq} & B::=E==E^{\prime },\\ F^{B} & B::=\neg B\;|B\wedge B^{\prime }\;|\;B\vee B^{\prime }, \end{array}\right.$$
*where $F^{Eq}=F_{LIn_{i}}\left(X⇝E\right)+F_{LIn_{i}}\left(X⇝E^{\prime }\right)$, $F^{B}=F_{LIn_{i}}\left(X\overset{B}{⤏}Y\right)+F_{LIn_{i}}\left(X\overset{B^{\prime }}{⤏}Y\right)$.*


**QR** 2 is used to analyze the leakage of the explicit flow. It is easy to follow that, when there is an explicit flow from *X* to *Y*, we regard this as all the information of *X* having been delivered to *Y* based on knowledge of $LIn_{i}$, i.e., $H(X|Y,LIn_{i})=0$.

**QR** 3 is used to analyze the leakage of the implicit flow, which includes the following three cases.

(1) For the basic boolean expressions $\left(E<E^{\prime }\right)$ or $\left(E>E^{\prime }\right)$, we consider the worst case in which the value of *B* and $E^{\prime }$ can be observed based on the knowledge of *Y* and *Z*. Then, attackers can deduce one more bit information about *X* in *E* at most after the service execution, which complies with 1−Bit rule in [20].

For example, for the following program in which *x* is the 5 bits high level input ranging from −16 to 15, *y* is the low level output,
$$\left\{\mathrm{state}\;s\right\}\mathrm{if}\left(x<z\right)\;\mathrm{then}\;y=0\;\mathrm{else}\;y=1\left\{\mathrm{state}\;s^{\prime }\right\}.$$
If attackers know the value of *z*, then he can deduce if the value of *x* is greater or less than *z* through the output value of *y*. Based on information theory, the entropy of *x* in state *s* and $s^{\prime }$ can be calculated as follows:$$\begin{array}{c} H(x_{s}|z_{s})=5,\\ H\left(x_{s}|z_{s},y_{s^{\prime }}\right)=P_{x<z}H\left(x_{s}|z_{s},y_{s^{\prime }}=0\right)+\\ \;\;\;\;\;\;\;\left(1-P_{x<z}\right)H\left(x_{s}|z_{s},y_{s^{\prime }}=1\right). \end{array}$$
In addition, we can also get that $H(x_{s}|z_{s},y_{s^{\prime }})$ is minimum when z=0, i.e., $Min\left(H\left(x_{s}|z_{s},y_{s^{\prime }}\right)\right)=\left(1/2\right)log\left(16\right)+\left(1/2\right)log\left(16\right)=4$. Then, the attacker can obtain one bit of information about *x* at most through this flow.

(2) For the equality expression $\left(E==E^{\prime }\right)$, it is a special case in which service providers may obtain all bits of *X* in *E* or $E^{\prime }$ when this expression is true. In this case, the leakage depends on how much information leaked from *X* to *E* and $E^{\prime }$, i.e., $F_{LIn_{i}}\left(X⇝E\right)+F_{LIn_{i}}\left(X⇝E^{\prime }\right)$.

(3) For the complex expressions (¬B), $\left(B\wedge B^{\prime }\right)$ and $\left(B\vee B^{\prime }\right)$, the leakage of *X* depends on the quantity of leakage on each condition *B* and $B^{\prime }$, i.e., $F_{LIn_{i}}\left(X\overset{B}{⤏}Y\right)+F_{LIn_{i}}\left(X\overset{B^{\prime }}{⤏}Y\right)$.

These quantifying rules are consistent with the rules in [20]. Based on the quantifying on explicit and implicit flows, we can calculate the overall leakage from *X* to *Y* through different flows by the following rules.

**QR** **4.**
*$\forall X\in HIn_{i}$ and $Y\in LOut_{i}$ satisfy X⇝Y, then we have*
(7)$$F_{LIn_{i}}\left(X⇝Y\right)=\sum_{\delta \left(X,Y\right)\in F_{i}}F_{LIn_{i}}\left(\delta \left(X,Y\right)\right).$$


For **QR** 4, we also consider the worst case in which the leakage of information through each flow is different. Then, the overall quantity on leakage from *X* to *Y* is the sum of the leakage in each flow δ(X,Y). Based on the above quantifying rules and Definition 1, we can derive the following theorem on information flow security in $s_{i}$.

**Theorem** **1.**
*$\forall X\in HIn_{i}$ in $s_{i}$, the flows in $s_{i}$ are K-secure if they satisfy that*
$$\sum_{Y\in LOut_{i}}F_{LIn_{i}}\left(X⇝Y\right)<K,\;0\leq K\leq k,$$
*where K is the security threshold.*


**Proof.** Based on the information entropy and the quantifying rules, it is easy to deduce that
$$F_{LIn_{i}}\left(X⇝LOut_{i}\right)\leq \sum_{Y\in LOut_{i}}F_{LIn_{i}}\left(X⇝Y\right)<K.$$According to Definition 1, the flows in $s_{i}$ are secure. □

### 3.3. Quantifying the Information Flow in the Service Chain

In our threat model, different service providers may collude together to analyze s user’s private data. It means that different providers may share their knowledge on the low-level inputs and outputs during the service composition, which causes more leakage on a user’s private data. In order to quantify the additional leakage of private data across different services, we design the quantifying rules based on the analysis of the inter-service flows.

For service chain $S_{ch}=\langle s_{0},s_{1},s_{2},\ldots ,s_{n}\rangle$ where $In_{ch}={⋃}_{0\leq i\leq n}In_{i}=In_{0,1,\ldots ,n}$ and $Out_{ch}={⋃}_{0\leq i\leq n}Out_{i}=Out_{0,1,\ldots ,n}$, the inter-service flows may occur between the inputs and outputs across multiple services, which is shown as Figure 3 and Figure 4.

Adjacent-service flow is the basic inter-service flow, which occurs because of the transmission on the intermediate result between the outputs and inputs across the adjacent services, such as the inter-service flow between $Out_{i,2}$ and $In_{i+1,2}$. Based on the adjacent-service flows, more inter-service flows occur due to the transitivity of the information flow, such as the inter-service flow between $In_{i,n}$ and $Out_{j,1}$, 0≤i<j. Therefore, we can formally define the inter-service flows as follows [7]:

**Definition** **3.**
*$\forall X\in In_{i}$ and $\forall Y\in Out_{j}$ where 0≤i<j≤n, there are following two cases.*

*(1) j=i+1: $\exists W_{1}\in Out_{i}$, $W_{2}\in In_{j}$, $W_{1}\to W_{2}$ that satisfy $X⇝W_{1}$ and $W_{2}⇝Y$, then $X⇝W_{2}$ and X⇝Y.*

*(2) j>i+1: $\exists W\in In_{l}\cup Out_{l}$, i<l<j that satisfy X⇝W and W⇝Y, then X⇝Y.*


Based on Definition 3, we can obtain all the inter-service flows. Here, we define $F_{ch}=\left\{X⇝Y|X\in In_{i},Y\in Out_{j},0\leq i<j\leq n\right\}$. According to the composition structure of service chain model, the intermediate result is the only method that passes the value of input source across multiple services. Then, we can obtain the following proposition.

**Proposition** **1.**
*$\forall X\in In_{i}$ and $\forall Y\in Out_{j}$, 0≤i<j≤n, if X⇝Y, $\exists W\in In_{j}$ satisfies that X⇝W and W⇝Y.*


On the basis of Proposition 1, for each inter-service flow X⇝Y, its leakage from *X* to *Y* depends on the quantity of information that *X* passes to *W* and how much information is leaked through the intra-service flow W⇝Y. Then, we use $F_{LIn_{i,i+1,\ldots ,j}}{\left(X⇝Y\right)}_{W}$ to represent the additional leakage of *X* to *Y* through *W*, which can be calculated based on the following rule.

**QR** **5.**
*$\forall X\in HIn_{i}$, $Y\in LOut_{j}$ and $W\in In_{j}$ satisfy X⇝W and W⇝Y, then we have*
(8)$$F_{LIn_{i,i+1,\ldots ,j}}{\left(X⇝Y\right)}_{W}=\left\{\begin{array}{cc} 0, & W\in {\mathrm{LIn}}_{j},\\ F_{LIn_{j}}\left(W⇝Y\right), & W\in {\mathrm{HIn}}_{j}. \end{array}\right.$$


For the inter-service flow X⇝Y through *W*, there are two cases to consider in **QR** 5.

(1) $W\in LIn_{j}$: Because $W\in LIn_{j}$, the information of *X* is leaked through *W*. Then, service providers can not obtain additional information about *X* through the flow W⇝Y. For this type of flow, $F_{LIn_{i,i+1,\ldots ,j}}{\left(X⇝Y\right)}_{W}=0$.

(2) $W\in HIn_{j}$: During the service composition, it is considered secure that private data are delivered between high-level sources and sinks. Thus, the explicit flows usually occur between the high-level inputs and outputs. In this case, we also consider the worst assumption that all the information of *X* is delivered to *W* based on **QR** 2. Because $W\in HIn_{j}$, the information of *X* can not be leaked through *W*. The leakage of *X* depends on how much information of *W* leaks through the flow W⇝Y. In addition, we assume that the leakage from *X* to *Y* is different from the previous flows. Then, we can obtain that $F_{LIn_{i,i+1,\ldots ,j}}{\left(X⇝Y\right)}_{W}=F_{LIn_{j}}\left(W⇝Y\right)$.

In addition, for each inter-service flow X⇝Y, the information of *X* may leak to *Y* through different *W*. Then, we can deduce the following lemma.

**Lemma** **1.**
*$\forall X\in Hin_{i}$ and $Y\in LOut_{j}$, 0≤i<j≤n, satisfy X⇝Y, then*
$$F_{LIn_{i,i+1,\ldots ,j}}\left(X⇝Y\right)\leq \sum_{W\in HIn_{j}}F_{LIn_{j}}\left(W⇝Y\right),$$
*where W satisfies X⇝W and W⇝Y.*


**Proof.** According to the above analysis and the service chain model, we can deduce that
$$F_{LIn_{i,i+1,\ldots ,j}}\left(X⇝Y\right)\leq \sum_{W\in In_{j}}F_{LIn_{i,i+1,\ldots ,j}}{\left(X⇝Y\right)}_{W}=\sum_{W\in HIn_{j}}F_{LIn_{j}}\left(W⇝Y\right)$$
lemma is proved. □

Based on Lemma 1, we can obtain that

**Lemma** **2.**
*In a service chain $S_{ch}=\left\{s_{0},s_{1},\ldots ,s_{n}\right\}$, $\forall X\in Hin_{i}$ and $\forall Y\in LOut_{ch}$ satisfy X⇝Y, then*
$$F_{LIn_{i,i+1,\ldots ,n}}\left(X⇝LOut_{ch}\right)\leq L{\left(X\right)}_{i}+L{\left(X\right)}_{i+1,\ldots ,n},$$
*where $L{\left(X\right)}_{i}$ is the leakage of X to $\forall Y\in LOut_{i}$ in service $s_{i}$, namely,*
(9)$$L{\left(X\right)}_{i}=\sum_{Y\in LOut_{i}}F_{LIn_{i}}\left(X⇝Y\right),$$
*and $L{\left(X\right)}_{i+1,\ldots ,n}$ is the additional leakage of X to $\forall Y\in LOut_{j},i<j\leq n$ in following services $s_{i+1},\ldots ,s_{j}$, namely,*
(10)$$L{\left(X\right)}_{i+1,\ldots ,n}=\sum_{j=i+1}^{n}\sum_{Y\in LOut_{j}}\sum_{W\in HIn_{j}}F_{LIn_{j}}\left(W⇝Y\right),$$
*and W satisfies X⇝W and W⇝Y in Equation (Equation 10).*


**Proof.** The proof is shown in Appendix A. □

Based on Lemma 2, we can derive the following information flow security theorem.

**Theorem** **2.**
*For a service chain $s_{ch}=\left\{s_{0},s_{1},\ldots ,s_{n}\right\}$, the information flows are K-secure if each service component $s_{j},0\leq j\leq n$, satisfies the following two conditions:*

*(1) Flows in each service component $s_{j}$ are secure.*

*(2) $\forall X\in HIn_{i}$, 0≤i<j; it satisfies that*
(11)$$L{\left(X\right)}_{i,\ldots ,j-1}+L{\left(X\right)}_{j}\leq K,$$
*where K is the security threshold. $L{\left(X\right)}_{i,\ldots ,j-1}$ is the overall leakage of X from $s_{i}$ to $s_{j-1}$, namely,*
(12)$$L{\left(X\right)}_{i,\ldots ,j-1}=L{\left(X\right)}_{i}+L{\left(X\right)}_{i+1,\ldots ,j-1}$$
*and $L{\left(X\right)}_{j}$ is the leakage of X in $s_{j}$, namely,*
(13)$$L{\left(X\right)}_{j}=\sum_{Y\in LOut_{j}}\sum_{W\in HIn_{j}}F_{LIn_{j}}\left(W⇝Y\right)$$
*and W satisfies X⇝W and W⇝Y in Equation (Equation 13).*


Theorem 2 can be proved based on Lemma 2 and Definition 1. The security constraints on each service are given in Theorems 1 and 2, which makes a basis for the decentralized evaluation in mobile computing environment. Each service requires that the leakage of high level data through the intra and inter flows can not exceed the threshold *K*.

## 4. Distributed Quantitative Information Flow Evaluation for Service Composition in a Mobile Computing Environment

In MSS, services may be composed together to accomplish a user’s complex service requirement. For a service chain $S_{ch}=\left\{s_{0},s_{1},s_{2},\ldots ,s_{n}\right\}$, there are several candidate service components with the same functions but different providers for each service step $s_{i}$. In order to efficiently evaluate the leakage for the service composition in a mobile computing environment, we propose a distributed quantitative information flow evaluation approach based on Theorems 1 and 2.

By referring to Figure 1, candidate services and security authorities will be involved in the evaluation procedure. The procedure includes two phases, i.e., intra-service evaluation and inter-service evaluation. First, each candidate service is evaluated by its local SA, and SA generates a security certificate for the following inter-service evaluation. When these candidate services are going to be composed together, the inter-service evaluation process will be executed for the evaluation on leakage by inter-service flows.

### 4.1. Intra-Service Evaluation

The intra-service evaluation is executed by SA before the service composition. SA evaluates each candidate service $s_{i}$ based on the quantifying rules and Theorem 1, and generates security certificates $Ce_{i}$ for secure ones. This phase can be executed in an offline way to reduce the evaluation cost during the composition.

During the intra-service evaluation, SA first obtains the PDG of $s_{i}$, then computes the quantity of leakage from $\forall X\in HIn_{i}$ to $\forall Y\in LOut_{i}$ based on the above QRs. After that, SA validates the flow in $s_{i}$. For secure services, a certificate $Ce_{i}$ specifying the quantity of leakage on each high-level inputs $L{\left(X\right)}_{i}$ is generated for the following evaluation. Insecure ones without certificates are not allowed to be composed during the service composition. The intra-service evaluation procedure is presented as Algorithm 1.

In the computation of the leakage on X⇝Y, we record the value in the certificate which can be used in the inter-service evaluation phase. It can avoid the repeated work on quantifying leakage in a same service component. At the end of the procedure, we record the flows between high level inputs and outputs in certificate instead of computing its leakage. It is based on our worst assumption that information has been passed to high level outputs if there is a flow, which usually happens. In the meantime, it can save lots of efforts on computation of leakage during inter-service evaluation without loss on security.

For a clear description on our intra-service evaluation algorithm, consider the following service’s program:

**public static int**Compare(**int**hin, int lin){  **int** hout, lout;  hout=hin;  lout=−1;  **if**(hin>lin)   lout=0;  **else**   lout=1;
}


In the above example, hin and hout are high-level inputs and outputs while lin and lout are low-level ones. First, the code needs to be sent to SA. Then, SA constructs the PDG of ’Compare’ service and obtains the intra-service flow set $F_{i}=\left\{hin\to hout,hin\overset{hin>lin}{⤏}lout,lin\overset{hin>lin}{⤏}lout\right\}$. After that, we compute the leakage from hin to lout through the flow $hin\overset{hin>lin}{⤏}lout$ based on **QR** 3. The leakage is validated according to security threshold *K*. If it is considered secure, the leakage of hin through each flow, current overall leakage of hin and the intra-service flows between hin and hout will be recorded in certificate $Ce_{i}$ for the inter-service evaluation. Finally, certificate $Ce_{i}$ is signed by SA for the protection against manipulation.


**Algorithm 1**
Intra_Eval()
**Input:**$s_{i}$, *K***Output:****True** or **False**, $Ce_{i}$.
1:generate the $s_{i}$’s PDG and obtain $F_{i}$2:**for** each $X\in HIn_{i}$
**do**3: **for** each $Y\in LOut_{i}$
**do**4:  **for** each $\delta \left(X,Y\right)\in F_{i}$
**do**5:   compute $F_{LIn_{i}}\left(\delta \left(X,Y\right)\right)$ based on **QR** 2 and **QR** 36:   $F_{LIn_{i}}\left(X⇝Y\right)$ = $F_{LIn_{i}}\left(X⇝Y\right)$ + $F_{LIn_{i}}\left(\delta \left(X,Y\right)\right)$7:  **end for**8:  record the $F_{LIn_{i}}\left(X⇝Y\right)$ into service certificate $Ce_{i}$9:  $L{\left(X\right)}_{i}$ = $L{\left(X\right)}_{i}$+ $F_{LIn_{i}}\left(X⇝Y\right)$10: **end for**11: **if**
$L{\left(X\right)}_{i}\geq K$
**then**12:  **return False**13: **end if**14: record the $L{\left(X\right)}_{i}$ into service certificate $Ce_{i}$15: **for** each $Y\in HOut_{i}$
**do**16:  **if**
$\exists \delta \left(X,Y\right)\in F_{i}$
**then**17:   record the flow from *X* to *Y* into service certificate $Ce_{i}$18:  **end if**19: **end for**20:
**end for**
21:signature($Ce_{i}$,SA)22:
**return True**



### 4.2. Inter-Service Evaluation

Inter-Service evaluation is a vital phase to evaluate the leakage of high level data during service composition. In this phase, $s_{i}$ firstly retrieves current leakage on high level data $L{\left(X\right)}_{0,1,\ldots ,i}$ and their inter flows $F_{0,1,\ldots ,i}$. Then, $s_{i}$ requires $s_{i+1}$’s intra flow and leakage through the certificate $Ce_{i+1}$, and it updates the inter-service flow set and evaluates the candidate service $s_{i+1}$ according to Theorem 2. The inter-Service evaluation procedure is shown as Algorithm 2.

During the evaluation on candidate service $s_{i+1}$, the additional leakage $L{\left(X\right)}_{i+1}$ is first calculated based on **QR** 5 and Lemma 1. After that, the overall leakage of each high-level input, $L{\left(X\right)}_{0,1,\ldots ,i+1}$, is computed and validated. If the overall leakage on any high-level input exceeds security threshold *K*, it means that this candidate service $s_{i+1}$ is not secure for composition.


**Algorithm 2**
Inter_Eval()
**Input:**$s_{i+1}$, *K*, $L{\left(X\right)}_{0,1,\ldots ,i}$, $F_{0,1,\ldots ,i}$**Output: True** or **False**, L(X)0,1,…,i+1, $F_{0,1,\ldots ,i+1}$.
1:retrieve cert $Ce_{i+1}$2:update the flows $F_{0,1,\ldots ,i+1}$ based $F_{0,1,\ldots ,i}$ and $Ce_{i+1}$3:**for** each $X\in HIn_{j},j<i+1$
**do**4: **for** each $Y\in LOut_{i+1}$
**do**5:  **for** each $W\in HIn_{i+1}$
**do**6:   **if**
$\left(X⇝W\right)\wedge \left(W⇝Y\right)\in F_{0,1,\ldots ,i+1}$
**then**7:    get the leakage $F_{LIn_{i+1}}\left(W⇝Y\right)$ from the certificate $Ce_{i+1}$8:    $L{\left(X\right)}_{i+1}$ = $L{\left(X\right)}_{i+1}$ + $F_{LIn_{i+1}}\left(W⇝Y\right)$9:   **end if**10:  **end for**11: **end for**12: $L{\left(X\right)}_{0,1,\ldots ,i+1}$ = $L{\left(X\right)}_{0,1,\ldots ,i}$ + $L{\left(X\right)}_{i+1}$13: **if**
$L{\left(X\right)}_{0,1,\ldots ,i+1}\geq K$
**then**14:  **return False**15: **end if**16:
**end for**
17:
**return True**



### 4.3. Distributed Quantitative Information Flow Evaluation Algorithm for the Service Composition in Mobile Computing Environments

Based on the intra-service and inter-service evaluation procedure, we propose a distributed quantitative information flow evaluation algorithm for service composition across multiple mobile devices. The evaluation algorithm on each high level input is presented as Algorithm 3.


**Algorithm 3**
Eval_SC()
**Input:**$s_{i}$, $s_{i+1}$*K*
**Output:**
L(X)0,1,…,n

1:wait **start_message**2:**if***X*∈$HIn_{i}$**then**3: \\ Initiate the *X*’s leakage in its first service4: $L{\left(X\right)}_{i,i+1,\ldots ,n}=L{\left(X\right)}_{i}$
5: $F_{i,i+1,\ldots ,n}=F_{i}$6: send **start_message** to $s_{i}+1$’s SA7:
**else**
8: get $L{\left(X\right)}_{0,\ldots ,i}$ and $F_{0,\ldots ,i}$ from **start_message**9: **if**
Inter_Eval($s_{i+1}$,*K*,$L{\left(X\right)}_{0,\ldots ,i}$, $F_{0,\ldots ,i}$)=**Fail then**10:  send **fail_message** to the user11: **else**12:  **if** i=n **then**13:   send **success_message** to the user14:  **else**15:   send **start_message** to $s_{i+1}$16:  **end if**17: **end if**18:
**end if**



The algorithm is deployed on each service node in a mobile computing environment. Then, it works in a step-by-step way through the cooperation among multiple services in different mobile devices. For each possible service chain, a user sends a start message to the first service $s_{0}$ to start the evaluation procedure. During each step evaluation, each candidate service $s_{i+1}$ is evaluated by its predecessor $s_{i}$. If it returns true, $s_{i}$ will send a start message with current leakage and flows to its successors to continue the evaluation procedure. Otherwise, $s_{i}$ will send a failure message to a user to check whether this service chain is not secure, and the evaluation on this chain will stop. In addition, for each high-level input in the service chain, it needs to be initiated in its first service based on the certificate. When the final service $s_{n}$ passes the evaluation, then it will send a success message with the overall leakage $L{\left(X\right)}_{0,1,\ldots ,n}$ on each high level input to user. The leakage can be used as a security criterion on different candidate service chains.

## 5. Experiments and Evaluations

The information flow security can be ensured by Theorem 2, and the security proof and analysis are shown in Appendix A. The basic comparisons of related approaches are shown in Table 1.

According to Table 1, traditional approaches validate the information flow across multiple services based on non-interference, i.e., qualitative verification. Comparing with quantitative approaches for a program, our approach supports the distributed quantifying on the flows across multiple services, which is more appropriate for the service composition in a mobile computing environment. Although we refer to the rules in [20], these rules are used to quantify the overall leakage of high-level inputs instead of each flow’s leakage, which is more suitable for the centralized evaluation on a single program.

We also implemented our approach in Huawei mobile phones and ran the evaluation procedure in a WLAN network with the speed of 150 Mbps. The basic configuration is shown in Table 2.

In order to evaluate the performance of our approach, we construct a data set of android applications. These applications support two security levels, i.e., H (High) and L (Low). Each application has two high-level inputs, two low-level inputs, two high-level outputs and two low-level outputs. The flows between different inputs and outputs in each application are randomly generated. These applications can be regarded as basic services in mobile computing environments, which can be composed together as a composite service by network communication. For the composite service, the number of service step $N_{s}$ is from 1 to 10. The number of candidate service $N_{c}$ for each step is also from 1 to 10. $N_{s}$ means this composite service is composed by $N_{s}$ types of applications. $N_{c}$ means there are $N_{c}$ applications having similar functions but different implementations for each type service. In our experiments, we focus on the evaluations on service availability, time cost and energy cost.

**(1) Service Availability**: we use success number and success rate to evaluate the availability of the composite service. Success number $N_{suc}$ is the number of the composite services that successfully pass the validation. In addition, success rate $R_{suc}$ is the percentage of successful composite services in all possible ones, which can be calculated as the following equation:(14)$$R_{suc}=\frac{N_{suc}}{N_{all}},$$
where $N_{all}$ is the number of all possible composite services composed of different candidate applications in our data set.

In this test, we execute the quantitative approaches with different security threshold (*K* = 8, 16, 32) and qualitative approaches 100 times separately. Figure 5 shows the average success number and success rate on service composition with different approaches.

Figure 5a shows the variation on average success number of service composition with fixed service steps ($N_{s}=5$) but different number of candidate services $N_{c}$. With the increase in $N_{c}$, the number is rising because it is easier to be successfully composed with more candidate services. Figure 5b shows the variation on average success rate of service composition with fixed candidate services ($N_{c}=5$) but a different number of service steps $N_{s}$. With the increase in $N_{s}$, the rate is declining because it is harder to find an appropriate service that can satisfy the security constraints.

Throughout both figures, the quantitative approaches have better performance compared to the qualitative approach. Especially for the success rate when $N_{s}=10$ in Figure 5b, few services could pass the qualitative validation which may cause failure on service composition. On the contrary, the success rate is still high in quantitative approaches. It indicates that the performance of the quantitative approach is apparently superior to that of the qualitative approach. Moreover, it is easier to be successful with a higher security threshold.

**(2) Time Cost**: we focus on the time cost on different types of information flow validation approaches. The time cost mainly includes two types of costs, i.e., computation and communication. Figure 6a shows the overall time cost including computation and communication on qualitative and quantitative approaches. With the increase in the number of candidate services, the time cost is rising because of the increase in the complexity and the number of possible service chains. It also costs more time due to the additional computations and communications in quantitative evaluation approaches. However, we can minimize the cost by precomputation on the quantity of leakage in each candidate service.

Figure 6b shows the average computation cost on each mobile device involved in the distributed and centralized quantitative approaches. We use the time cost on computation to represent the computation cost in this test. Instead of executing all of the evaluation work on a single device, the distributed way coordinates all participants to accomplish the evaluation together, which provides a better balance on the computation cost on mobile devices.

**(3) Energy Cost on User’s device**: During the evaluation, the energy cost is mainly caused by the computation and the communication cost on a user’s device. In our distributed approach, the user starts the evaluation with little computation cost. The computation cost is also evaluated in the above ’time cost’ test. Thus, we focus on the communication cost on the user’s mobile device in this test.

Figure 7 shows the communication cost on the user’s mobile device in the distributed and centralized evaluation approaches. For the centralized evaluation approach, all candidate services will be evaluated by user’s device. For our distributed approach, a user’s device only needs to send the request to its following services and receives the final results from the last-step services. Therefore, the communication cost in a centralized approach is higher than that in our distributed approach. By combining the evaluation on the computation cost, they indicate that the energy cost on a user’s device can be reduced by our distributed approach.

Based on the above experiments, the results show that our approach can provide better service availability with a small increase in time cost, provide a better load balance on computation and reduce the overload on the users’ mobile phones effectively.

## 6. Conclusions

Strict qualitative disciplines decrease the availability of the composite service and may cause a high failure rate on service composition. In this paper, we propose a distributed quantitative information flow evaluation approach for secure service composition in mobile computing environments. Our approach first evaluates the intra-service leakage between different inputs and outputs in each service, and then ensures the inter-service flow security based on the constraints specified in Theorem 2. Our framework and approach works in a distributed way which is quite suitable for the evaluation executed by energy-limited devices in mobile computing environments. Through experiments and evaluations, the results show that our approach can improve the service availability effectively and provide a better load balance on each device. 

## Figures and Tables

**Figure 1 entropy-21-00753-f001:**
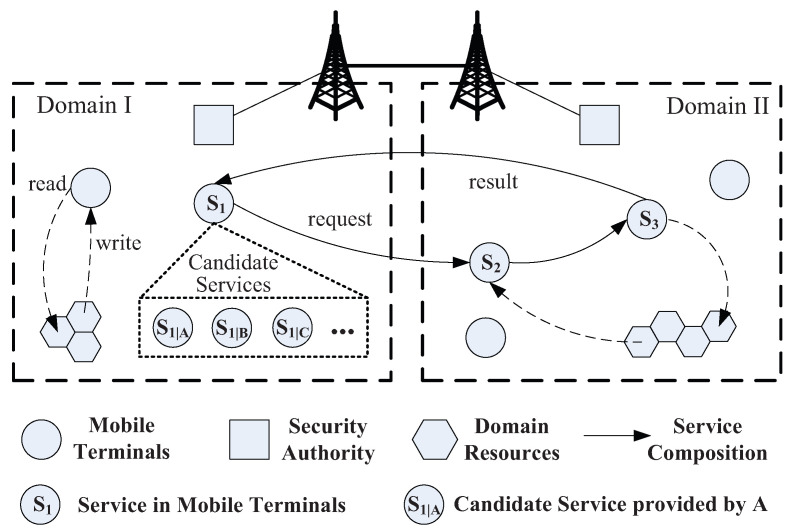
Mobile service system.

**Figure 2 entropy-21-00753-f002:**
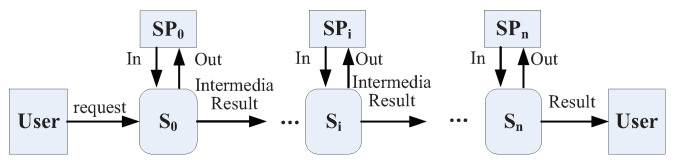
Service chain model in mobile service system.

**Figure 3 entropy-21-00753-f003:**
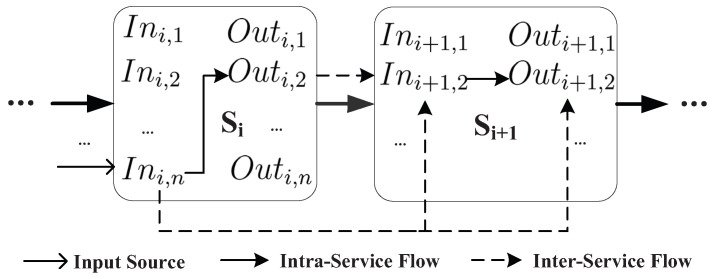
Information flow between adjacent services.

**Figure 4 entropy-21-00753-f004:**
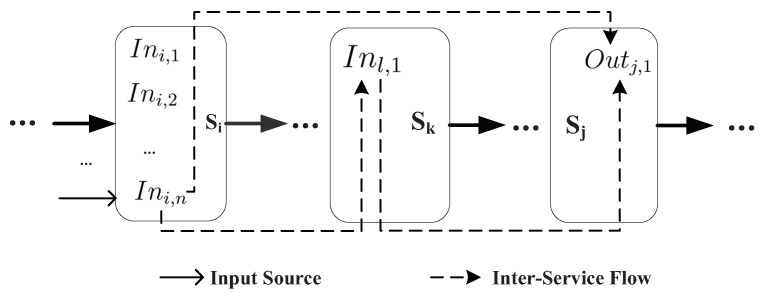
Information flow across multiple services.

**Figure 5 entropy-21-00753-f005:**
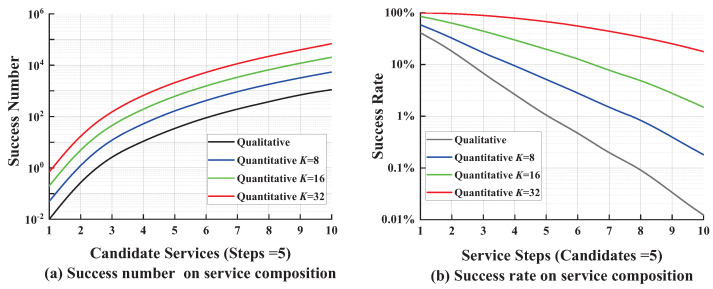
Success number and success rate on service composition.

**Figure 6 entropy-21-00753-f006:**
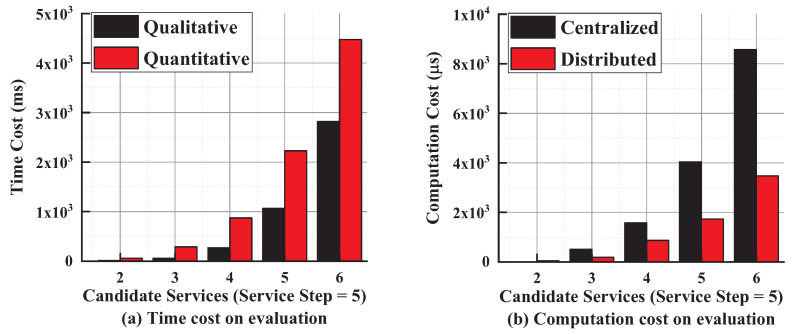
Time cost on information flow evaluation.

**Figure 7 entropy-21-00753-f007:**
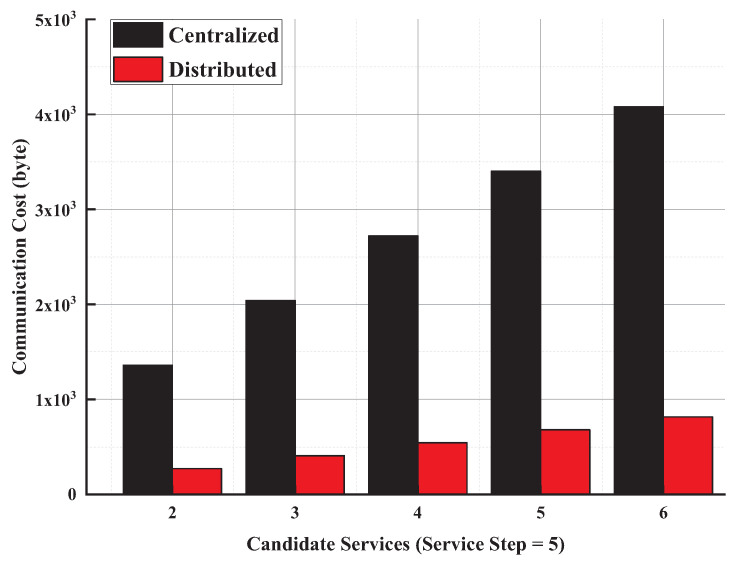
Communication cost on information flow evaluation.

**Table 1 entropy-21-00753-t001:** Basic comparison.

	Approach	Mode	Service Composition
Our Approach	Quantitative	Distributed	√
She et al. [13,14]	Qualitative	Centralized	√
Xi et al. [25]	Qualitative	Distributed	√
Clark et al. [20,24]	Quantitative	Centralized	×
Smith et al. [22]	Quantitative	Centralized	×

**Table 2 entropy-21-00753-t002:** Configuration.

**Mobile Environment**	
Network Type	WLAN
Network Speed	150 Mbps
Mobile Mode	random walk
Mobile Devices	Huawei nova 3
Device’s CPU and RAM	2.8 GHz, 6 G
Mobile Device’s Operation System	Android 9.0
**Data Set**	
Service Step	1–10
Candidate Number	1–10
Security Level	H, L
High Level Input and Output	2, 2
Low Level Input and Output	2, 2
Flows between Input and Output	randomly generated

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
