# Peer review of "Secure Service Composition with Quantitative Information Flow Evaluation in Mobile Computing Environments"

_entropy, 2019, doi:10.3390/e21080753_

Round 1

Reviewer 1 Report

Summary of paper:

This paper presents an approach to compute quantitative flows on a chain of devices implementing mobile services. The paper presents a system model for mobile services, and defines the attacker model to be service providers who can colude with each other, and observe low inputs and low outputs. Information flow is defined as mutual information between high inputs and low outputs, given low inputs, and is estimated as a worst-case scenario as the entropy of a uniform distribution on inputs. A method for computing leakage in a distributed form is proposed, and experiments are performed to show the method allows for more service availability than qualitative approaches (when a threshold of leakage is tolerated), with only a moderate cost on computing time.

Evaluation:

This paper presents some interesting ideas, and the experimental results suggest they may be promising. However, the paper is not ready for publication. The presentation is very confusing, notation is not used consistently, and the proof of the main result is so cryptic that I could not check it to the end.

Regarding notation, if you are measuring I(X;Y | Z) why use the notation F_Z(X -> Y)? It makes it much harder to understand the proofs (specially uses of chain rules), and the use of subscripts is very heavy. Unless the authors can make a convincing argument about the advantages of their notation, they should stick to the common notation already in use. Moreover, the authors seem inconsistent in calling a same set of variables as sometimes Y, sometimes Out, sometimes LOut. As another example of how notation can be confusing, in QR2 and QR3 X is either of type In or of type HIn. What's the difference between the two cases? I don't have confidence in whether Hin <> In, or if it is just inconsistent notation.

In terms of presentation, why aren't service providers part of the model (defined in Section 2.1)? Moreover, the use of types is sloppy. For instance, in QR 1 (on page 4), it would be more clear to define In to be the set of k-bit variables, and just write X \in In. This happens in several other places. Also, the expression "there is" is often misused, creating situations in which it is hard to understand what the authors mean. In the same QR 1, for instance, where it reads "there is" I understand that what is meant is "it is the case that".

Also regarding presentation, the experiment section is also unclear. "Success amount" is not defined (I assume it is the number of instances in which service could be provided), and it is not clear whether it is an absolute number or an average (if it is an average, it is an average of what?). It is not clear how "computational cost" is different from "time".

A final comment about presentation is that an Appendix usually comes after references, and not before.

On a technical side, why isn't the leakage of page 6, line 180 (for expression E == E') defined as a sum, instead of a maximum value? I.e., max(F_{ziin}(X->E),F_{ziin}(X->E')) instead of F_{ziin}(X->E) + F_{ziin}(X->E')? The same applies to def. of page 7 line 182 (for boolean expressions).

My main concern with the paper, however, is on the validity of some results, as the proofs are either missing or blurred by the heavy, inconsistent notation. For instance, Equation (8) is not "easy" at all to see (on page 7, line 204-205). More seriously, the proof of Lemma 1 in the appendix is quite cryptic. Equation (A1) is not at all easy to see, even because it is very hard to parse. The left-hand side of the equation is just a formula for mutual information, and I assume that some chain rule is applied here. I suggest the authors to clarify (maybe using the notation of mutual information) what is going on. Also in Equation (A2) the range of the summation on Y is not stated. Finally, on line 318 some notation is introduced for the "quantity X passed to Y through W", but with no definition. It is very hard to check whether the proof is correct at all with all this confusion with notation, so I gave up at this point.

Because of the exposed above, I can not have confidence in the theoretical results of the paper until presentation is substantially cleaned up so I can check them properly. I have made several suggestions in the list of minor comments below, but the paper would need much more work than that.

Minor comments:

Page 1:

- "tasks for satisfying" -> "tasks to satisfy"?

- "high speed processor" -> "high speed processors"?

- "architecture(SOA)" -> leave space between word and acronym (happens elsewhere in the paper too)

- "if there is one service component contains" -> "if one service component contains"

Page 3:

- "and start the service procedure" -> "and starts the service procedure"

- "Then each service s_i receives" -> "Then each service s_i, for 0 < i < n, receives"

- "who execute" -> "who executes" (ln 102)

- Why can't service providers observe the low-level inputs and outputs during execution? Does that make any difference?

Page 4:

- "Security Authority are" -> check plural vs singular

- The definition of conditional entropy on line 102 is wrong, the p(x) on the summation should be p(x |y).

Page 5:

- The rule P::=P;C is missing a base case. Shouldn't it be P ::= P;C | C?

- The sentence of line 144, explaining explicit flows, is not clear. 

Page 6:

- When first introducing an acronym, spell it out, e.g., PDG.

Page 11:

- What is success amount? Is it a absolute number or an average? Average among what?

Page 12:

- What's "computational cost" in Figure 4? How's that different from time?

Reviewer 2 Report

The authors construct an algorithm to determine whether composition of services written in a simple imperative syntax maintain a bound on the overall amount of  information leakage about high input values (secrets) by observation of publicly observable outputs (low output values).

They adopt a worst-case analysis of information leakage (e.g. full information leakage for assignment instructions, 1 bit for branching instructions, assuming leakage from distinct flows add up and there is no overlap, measurements are exact, and the adversaries know everything except for high input and hight output values of each service component.)

The analysis of the inter-service leakage is not novel, as the authors cite a 2007 paper for it. The novelty and new contribution is perhaps in the inter-service leakage and then getting the overall leakage.

In terms of applicability, the problem is very important and practically relevant, however, I am not sure how much relevance is kept if the syntax is the simplistic one in the paper. The authors need to clarify that.

Also although the authors say their algorithm is "distributed", it is so in a limited sense: each service still relies on all the previous services having computed their leakages and flows. However, since the permission to continue the service chain stops as soon as the leakage threshold is violated, I think that is fine.

Another issue with the paper is that the evaluation has not made it clear on what example services the leakages are computed. Nothing at all is provided on what example programs were considered, what the range of high/low input/output variables were, etc.

Moreover, I was not clear what the x axis represents (there are confusing terms like service candidates and service steps).

I was also not clear about the role of the low inputs (Z_i) in the analysis. The authors need to clarify that.

Last but not least, there are MANY editing issues with the paper, mostly English grammatical mistakes. This had made it difficult to read the paper easily. Some of them are listed below:

1- Throughout the paper, there are either unnecessary "a" or "the" articles where there shouldn't be, or they are missing where it should have been. For instance, equations, figures, theorems, or definitions that are specified with a number do not need a "the" article (so instead of "according to the definition 1" it should just be "according to definition 1", and so on).

2- Throughout the paper, there should be a space between a word and a parentheses, or a bracket.  E.g., there should be a space before the parenthesis in "service-oriented architecture(SOA)" or before citation brackets.

3- The authors should avoid starting sentences with "And". They can either use "Moreover," or "Also," or nothing (usually, just removing the And works the best!)

4- Many verbs are not conjugated properly: for third person singular in present tense, the verb should be conjugated with an "s". For example, "Usually user is the owner of mobile device who execute the first service in the chain." should be "Usually the user is the owner of the mobile device which executes the first service in the chain." or "Security Authority are the trusted third parties who executes the security function honestly" should be "Security Authority are the trusted third parties who execute the security function honestly" (execute not executes), etc.

5- There are many typos, e.g. "is comply" -> complies, "related approached" -> related approaches, "We realize" -> we implemented, "may composed" -> may be composed, "can deduce the value" -> can deduce if the value, "Based on the definition on F_i" -> Based on the definition of F_i

6- Throughout the paper, the authors use the expression "there is" in presenting a fact or an observation, and almost always they mean "we have" (or just nothing, i,e, removing "there is" in some of the sentence is actually good enough). This should be fixed throughout the paper (there are probably over 10 incidents of this).

7- Some sentences need to be re-written for smoother reading, e.g. "For each service s_i, it is defined as a tuple ... " should simply be "Each service s_i is defined as a tuple ... " or even simpler: "Each service s_i is a tuple ..."

Reviewer 3 Report

Overall, it's a clearly written paper. But I do have a few concerns regarding the technical contents.

The problem can be better motivated. More concrete applications are expected to present the problem settings.

The authors should emphasize the novelty and significance of the contributions. Section 3 is based on some prior work, and it isn't clear to me the novel contributions in this quantitative information flow model. The algorithms in Section 4 need more explanations. Some examples would be helpful.

The evaluation is a bit weak to me. The workload and settings are unclear. The authors should clearly describe the test environment and settings.

Other minor issues:

Section 2.2 is not "attacker mode". Actually the objective and attacking approaches of the attackers are not clearly stated in this paper.

There are typos and grammatical errors throughout the paper. The authors should carefully proofread the submission. 

Round 2

Reviewer 1 Report

The authors have taken into consideration my comments from my previous review, and this version of the manuscript has substantially improved with respect to the previous one. I appreciate their effort to make notation uniform (even if non-standard), and the presentation of a more thorough explanation of experiments and the metrics used. The use of English has been substantially improved as well, even if it still has minor points for improvement (see minor comments below).

Regarding technical aspects of the paper, the separation of results into theorems and propositions makes the presentation simpler. As far as I checked the proofs I didn't find any obvious mistakes.

I believe the paper presents a interesting contribution to measuring information flows in composite systems, and I believe the current form of the manuscript is acceptable for publication after some light editing regarding the use of English.

Minor comments:

Page 3:

- ln 94: S_ch[7] (space missing before [7]).

Page 5:

- In Definition 1 the quantification over \mathbb{K} is imprecise.
It is more precise to say that flows are \mathbb{K}-secure, for some 0 <= \mathbb{K} <= k, if F_LIn(X -> LOut) < \mathbb{K}.

- ln 145 and elsewhere: when referring to a particular definition, theorem, lemma, it is usual to capitalize the first letter. E.g., "in Definition 1 we can see" instead of "in definition 1 we can see". Same for, e.g., page 11 ln 269: "Theorems 1 and 2", and page 14 ln 442: "Lemma 2".

Page 9:

- ln 9: "the only way that pass" -> "the only way that passes"

Page 14:

- ln 359: "and runs the evaluation" -> "and run the evaluation"
- ln 370: "the following equation." -> "the following equation:"

Reviewer 2 Report

The authors have more or less addressed my main concerns.

There is still room for improvement in terms of proof-reading. Some suggestions follow:

- page 2, line 57: "Besides, all candidate services must be reevaluated
even a small change of one service occurs" -> even "if" (missing if)

- page 2, lines 75 and 76: "could" -> "can" (twice)

- page 2, line 78: "Moreover, there are several candidate services who
can execute
the similar functions for a same service." -> "Moreover, there can be
several candidate services that can execute the same functions for a
given service." (main change is replacing "who" with "that")

- page 4, the line before eq. (4): "its" leakage -> "the" leakage

- page 4, last line: there should be a space before the parenthesis of (QR).

- page 5, line 132: "Then we can obtain the quantitative definition on
information flow security in a system shown as follow": remove "shown"

- Just a suggestion: (Regarding definition 1 and throughout the paper)
usually \mathbb{K} is used for "sets" not just a number, you could have
just used K for "security threshold". But this is just a notational
suggestion.

- There are still missing spaces before the brackets for citations
throughout the paper (e.g., lines 48, 55, 141, etc.). As far as I know,
there should always be a space between a word and a parenthesis or
bracket, but it could be an acceptable style too (?).

- In the syntax, I don't think there is a need for the "|C", because in "C", you alreay have a "skip|", but that's a minor point.  

- In Definition 2, instead of removing "there is", it seems to be better to say "then"

- page 6, line 160, "definition 2" -> "Definition 2"

- page 6, line 164, "for a high security assurance" -> for a "strong" security assurance

- page 6, definition QR2 (as well as in QR3, QR4, etc.): instead of writing "... satisfy X->Y, then", it will read better to say e.g.: "that satisfy X->Y, we have" or something like that.

- page 6, line 165, there should be a space after the comma of "i.e.,"

- page 6, line 168: has "been" delivered

- page 10, lines 250 and 259, page 11 line 300, page 12 line 323, and page 15 line 392, "theorem" -> "Theorem" (same thing with numbered lemmas)

- page 12, line 329: "Although we refer the rules in" -> "Although we refer to the rules in"

- page 13, line 331: runs -> ran

- page 13, line 332: as Table~2 -> in Table~2

- page 13, line 342: "success amount" -> success number (also in the next sentence)

- page 13, line 343: "amount" -> "number"

- Throughout the paper, I suggest instead of "qualitative" approach, use the clearer term of zero-leakage or "non-interference" that is used in the QIF literature.

- Why in Figure 5-a the number of successes is used but in Figure 5-b the rate? It seems that success rate is more appropriate for both.

- In figure 6-a, the bars show the combined costs of time and communication overhead, and in figure 6-b, only the computation costs. However, the centralised costs on 6-b are higher than those of 6-a (almost twice). Are they from different experiments? I would have expected the centralised cost to be less in 6-b than in 6-a, because it does not contain the communication cost. It is a bit confusing.

- page 17, first line: "Based on equation" -> "Based on equations"

- page 16 line 409 and page 17 line 416, instead of "In a conclusion" either say "In conclusion" or even better, say "Hence," or "Therefore"

Reviewer 3 Report

The authors addressed most of my concerns in this revision.

There are still English issues throughout the paper. Please carefully proofread it again. The font size of the 3 algorithms is a bit too small.
